# Relationship between Strength and Professional Quality of Life of Nurses Working Hospital Emergency Departments

**DOI:** 10.3390/ijerph20032052

**Published:** 2023-01-22

**Authors:** Alfredo Córdova-Martínez, Daniel Pérez-Valdecantos, Alberto Caballero-García, Hugo J. Bello, Enrique Roche, David Noriega-González

**Affiliations:** 1Department of Biochemistry, Molecular Biology and Physiology, Faculty of Health Sciences, GIR: “Physical Exercise and Aging”, Campus “Los Pajaritos”, University of Valladolid, 42004 Soria, Spain; 2Department of Anatomy and Radiology, Faculty of Health Sciences, GIR: “Physical Exercise and Aging”, Campus “Los Pajaritos”, University of Valladolid, 42004 Soria, Spain; 3Department of Mathematics, School of Forestry, Agronomy and Bioenergy Engineering, GIR: “Physical Exercise and Aging”, Campus “Los Pajaritos”, University of Valladolid, 42004 Soria, Spain; 4Institute of Bioengineering, Department of Applied Biology-Nutrition, University Miguel Hernández, 03202 Elche, Spain; 5Alicante Institute for Health and Biomedical Research (ISABIAL), 03010 Alicante, Spain; 6CIBEROBN (Physiopathology of Obesity and Nutrition CB12/03/30038) Carlos III Health Institute, 28029 Madrid, Spain; 7Department of Surgery, Ophthalmology, Otorhinolaryngology and Physiotherapy, Faculty of Medicine, Hospital Clínico Universitario de Valladolid, 47003 Valladolid, Spain

**Keywords:** cortisol, dynamometry, fatigue, lactate, quality of life, stress

## Abstract

Optimal quality of professional life can allow coping with stressful situations and avoiding fatigue. This is instrumental in jobs undergoing very demanding situations, such as nurses working in Emergency Departments. In this study, we investigated the relationship between functional and physiological indicators and the quality of professional life in nurses of Emergency Departments of two public hospitals. Fifty-nine professionals participated in the study. Cortisol (an indicator of stress) and post-test lactate (an indicator of fatigue) were analyzed in saliva and in the blood, respectively, at the beginning and at the end of the morning shift. The CVP-35 survey (Spanish version of the QPL-35 questionnaire) was used to assess the quality of professional life. Post-dynamometry circulating lactate levels increased significantly; meanwhile, salivary cortisol decreased in participants at the end of the working shift. Regarding dynamometry, no significant changes were observed. CVP-35/QPL-35 scores did not show significant changes in participants from both hospitals in all dimensions analyzed. In conclusion, nurses participating in the study displayed adaptation to stress accompanied by an acceptable level of professional quality of life. However, certain fatigue appears at the end of the working day. Altogether, results suggest that a better organization must be implemented at work, improving ergonomic, safe, and healthy conditions.

## 1. Introduction

A healthy physical condition leads to a dynamic state of energy and vitality to enjoy active leisure time and cope with unforeseen emergencies. This situation helps to manage fatigue while preventing at the same time hypokinetic diseases and favoring optimal intellectual capacity [1,2]. In addition to optimal physical status, Job satisfaction is one of the most important variables related to organizational behavior and quality of life. Quality of professional life (QPL) has been defined as the experience of well-being derived from the individual’s perceived balance between work challenges, intense and complex tasks, and available resources (psychological, organizational, and relational) to cope with professional demands [1]. The QPL depends on a variety of factors, including personal, family, and work. Meanwhile, personal and family factors can be modulated by individuals themselves and work-related variables used to be modified mainly by external actions coming from supervisor statements.

Particular jobs, such as in the case of healthcare professionals, have to face other factors, such as professional recognition, working conditions, and management style, that can be modified in the short and medium term [2,3,4]. There is a close relationship and balance between job satisfaction and performance [5]. In healthcare services, this perception of balance results in an improvement in the quality of services provided [6,7]. In addition, it has been observed that the perception of high responsibility is directly related to the degree of qualification of particular individuals [8].

Nursing, considered a multidisciplinary work dimension, is one of the positions mostly affected in the medium term by stress, job dissatisfaction, lower performance, and personal frustration. This particular situation leads to a significant imbalance when trying to conceal work and family life [9]. Work-related stress negatively affects the quality of life and health, resulting in moderate anxiety that makes it more difficult to restore sleep [10]. Among the tools available to assess the quality of life in health care professionals, the validated CVP-35 (Spanish version of the QPL-35 questionnaire) is the most reliable tool used in research [1,11].

On the other hand, muscle strength is currently considered a benchmark of health and well-being. In this context, muscle strength has been associated with a lower risk of mortality in the adult population [12]. Therefore, manual dynamometry is considered a standard test for the assessment of physical fitness. It has been proposed that handgrip strength (HS) may be considered a valuable predictor for predisposition to certain diseases, such as cardiometabolic pathologies and disability [13]. In addition, HS assessment is a simple, quick, and inexpensive method, allowing one to assess the isometric strength of fingers and forearms. The HS is a sensitive method to detect the loss of physiological muscle function, thereby being an indicator of fatigue in the long term [14]. From a physiological point of view, muscle strength was defined as the ability of the activated muscle to produce tension and may be related to the presence of an external resistance [15]. Minimum levels of muscle strength are required to perform current daily activities, maintain functional independence and participate in active recreational tasks displaying no stress or fatigue [16]. Therefore, the type of work is related to the degree of strength, and at the same time, grip strength is generally influenced by the individual’s health status and level of physical activity [17].

Increased muscle tension that results from sustained muscle contraction can result from physical stress (elevated physical demands, repetitive tasks, inadequate body position, etc.) as well as from psychosocial stress [18,19,20,21,22]. In a previous report, we studied the hormonal response (salivary cortisol and amylase) in health professionals working in Emergency Departments (ED). We observed that doctors and nurses in this health service used to undergo a certain level of stress [18]. In this context, three types of response to stress could be considered [23]: (a) psychological and biological silence (no anxiety and no cortisol increase), (b) elevated emotional reaction with biological silence (high anxiety levels with no cortisol increase) and (c) elevated emotional reaction together with a high biological response (high anxiety together with plasmatic cortisol). Both Stress and resulting anxiety affect the quality of life and health of professionals working in EDs, resulting in lower self-efficacy and sleep quality [19]. Particularly, emergency nurses are a segment directly affected by work stress due to the demanding physical and psychological tasks that they must undertake very often. For this reason, EDs at hospitals are considered places of high stress at work, mainly in the nursing segment. Therefore, the objective of the present report is to assess the QPL of nursing staff working at the ED of two public hospitals and its relationship with biological (cortisol and lactate) and functional (HS test) parameters as indicators of stress, fatigue, and muscle functionality, respectively.

## 2. Materials and Methods

### 2.1. Participants

An analytical, descriptive, and cross-sectional study was conducted in September 2019 during the morning shift in the ED of two Spanish public hospitals: “Hospital Clínico Universitario” of Valladolid (HCUV) and “Hospital Santa Bárbara” of Soria (HSBS). The project was approved by the Clinical Research Ethics Committee of the Burgos Health Area (Reference: CEIC 1984).

Participation in the study was voluntary, and no financial compensation was received. Written informed consent was obtained from each subject in accordance with CEIC protocols. Anonymity was preserved, and subjects were informed that they were free to withdraw the study at any time with no explanations. Inclusion criteria were: (a) good health, with no disabling mental or physical pathology, (b) absence of endocrine or any other pathologies, (c) age ≥18 years. Exclusion criteria were: (a) last medical treatment for more than 15 days in the last 30 days, (b) smoking or history of smoking in the last 5 years, (c) alcohol abuse or history of alcohol abuse in the last 5 years, (d) use of drugs that influence the hypothalamic-pituitary axis (HPA), such as glucocorticoids, steroids, beta-blockers, antidepressants, melatonin or any other psychoactive drug, (e) use of glucocorticoids in the last 3 months, (f) medically diagnosed neurological or psychiatric disease, (g) night shift work or activity in another institution. A total of 59 nurses (10 men and 49 women) were recruited for the study. The average age of the nurses participating in the study was in a very similar range, around 39 years (Table 1).

### 2.2. Determinations

Saliva samples for cortisol determination were obtained during a regular morning shift (from 8:00 AM or T1 to 16:00 PM or T2) with no extreme emergency events. In case a demanding emergency event occurred, saliva collection was postponed to an alternative day. Participants knew in advance (60 min) by a mobile phone call/message the time and date of the scheduled protocol to collect saliva samples by a specialized technician. Participants were instructed to avoid eating or smoking 60 min prior saliva sampling. Samples were obtained according to Salivette^®^ kit (Sarstedt International, Nombrecht, Germany) and kept on ice until they were frozen at −20 °C and stored in the laboratory for subsequent analysis. On the day of determination, samples were thawed, centrifuged at 3000 rpm for 5 min at 4 °C, and analyzed by ELISA cortisol immunoassay (SALV-2930 DRG, Marburg, Germany). Reference values were established according to literature and manufacturer information.

The QPL-35 survey was filled out at T1 to assess the quality of professional life at work. It is an anonymous, self-administered test collecting the workers’ perception of their QPL in a multidimensional way. Questionnaire was constructed on the basis of the demand control model formulated by Karasek [20], and the Spanish version (CVP-35) has been validated [4,5]. It consists of 35 questions on a scale of 1 to 10, with overlapping categories of “not at all” (values between 1 and 2), “somewhat” (values between 3, 4, and 5), “quite a lot” (values between 6, 7 and 8) and “a lot” (values between 9 and 10). The questions are grouped into 3 groups assessing 3 variables (Appendix A): (a) “managerial support” perceived by the worker (assessed by 13 questions: 2, 3, 4, 5, 10, 11, 14, 16, 20, 22, 23, 28 and 30), (b) “workloads” or job demands perceived by the worker (assessed by 11 questions: 1, 6, 7, 8, 17, 18, 19, 21, 24, 25 and 33), and (c) “intrinsic motivation” (assessed by 11 questions: 9, 12, 13, 15, 26, 27, 29, 31, 32, 34 and 35). In addition, the questionnaire incorporates a direct question on the perception of QPL only applied to professionals with managerial responsibilities: “I feel my quality of professional life…”. The answer is according to the above-mentioned scale (from 1 to 10).

Dynamometry was performed after cortisol sampling with an instrument to measure hand grip strength (Jamar Hydraulic Hand Dynamometer Preston, Jackson, MO, USA). Measurement were performed using the dominant hand and recording the highest from 3 attempts. An interval of 60 s between attempts was applied to avoid muscle fatigue. To perform the test, each participant should be in a standing position, shoulder adducted and neutrally rotated, elbow flexed at 90°, wrist in neutral position, and handle held in a cylindrical grip with the instrument clock facing the evaluator. Participants must perform the grip with the maximum possible strength by means of a rapid but continuous impulse until maximal strength is reached. A demonstration of dynamometer use is shown to participants before performing the hand grip test. Results are expressed in kg.

A blood drop was obtained after handgrip test. Circulating lactate was analyzed with a portable automatic analyzer “Lactate Scout 4” (SENSLAB GmbH, Leipzig, Germany).

### 2.3. Statistics

Data were analyzed with the software package R, R-Studio, and Python (Pandas, Numpy, Scipy). The quantitative variables studied do not conform to the normal distribution, so the bootstrap method was used for data analysis, which was used to calculate 95% confidence intervals for the median. In order to study the differences between the groups, the Krustal-Wallis hypothesis test was used, which is used to determine whether the medians of the different groups are different and is considered a non-parametric alternative to the ANOVA method.

## 3. Results

The CVP-35 questionnaire was passed to participants by our research team in a previous report after 2 months working period [21]. It has been observed that nurses manifested a lower “Managerial Support” (dimension 1 in CVP-35) than Medicine Doctors. “Workload” (dimension 2 in CVP-35) perception was very similar between nurses and Medicine Doctors. However, “Intrinsic Motivation” (dimension 3 in CVP-35) was higher in nurses than in Medicine Doctors. Altogether, the overall perceived QPL is higher in nurses than in Medicine Doctors but still low. Therefore, a detailed analysis of these particular scores in nurses was achieved because they have to perform very demanding physical work, and the answers in dimension one seem to go in a different direction compared to the other dimensions. To this end, and in order to complete the obtained results from the CVP-35 questionnaire, biological (cortisol and lactate) and functional (HS test) determinations were performed.

Men presented significantly higher values in the HS test compared to women (Table 2). However, no significant differences were observed in HS tests comparing T1 vs. T2, although a tendency to decrease was noticed in women from both hospitals. In men, this tendency was less evident (Table 2), presenting a modest tendency to increase the HSBS and a tendency to decrease the HCUV. Lactate levels were measured after the HS test. Lactate increased significantly in professionals (men and women) from both hospitals when the HS test was performed at T2 compared to T1 (Table 2). Nevertheless, the lactate levels of all participants were below the anaerobic threshold.

Regarding cortisol levels, a decrease was observed through the working shift (T1 vs. T2) in men and women from both hospitals. However, there were significant differences between men and women HCUV at T1, with no differences at T2. On the other hand, the change % were in the same ranges in men and women and were lower in the HCUV than in the HSBS (Table 3).

Table 4 shows the results of the CVP-35 questionnaire that were very similar, with no significant differences in the nursing staff of both hospitals.

The analysis of the same dimensions of the CVP-35, separated by gender, indicated no significant differences (Figure 1). However, it is noteworthy that in both hospitals, men were reported to have a significantly higher quality of life at work than women.

Table 5 and Table 6 show correlations between different variables separated by gender. Correlations into the same time (T1 and T2) were compared. In this line, men displayed no correlation o very weak correlation between the different variables analyzed so far (Table 5). On the other hand, women only displayed a weak positive correlation comparing dynamometry with cortisol at T1 and T2 (Table 6). In addition, a weak positive correlation was noticed between lactate and cortisol at T1, which becomes a negative correlation between the same variables at T2 (Table 6). No differences were observed comparing hospitals.

## 4. Discussion

Nurses working in EDs have to face more difficult working conditions due to the job stress derived from demanding situations in patients’ care, together with organizational demands that may constrain decision-making. In this line, nurses from EDs report frequent, sustained pressure periods and physically demanding situations, with less authority to make decisions, less adequate work procedures, and fewer rewards than the rest of the nurses from other hospital departments [22]. This problem affects the professional and working organization. Therefore, the satisfaction and QPL of these professionals have a direct impact on the quality of care to patients [21].

Activities performed daily in EDs require a certain level of strength, so assessment by dynamometry can provide information about the functional and fatigue status of the nurses [23]. Strength was assessed using the HS test, which requires little training and only takes a few minutes. In this line, Torres et al. [24] conducted a study analyzing HS and comparing it to different dynamometric techniques. They concluded that the HS test was the best and simplest, giving better information than others. In addition, Roberts et al. [25] performed an intervention in elders indicating that HS is able to reflect the overall health condition of the individual. Moreover, a recent systematic review [26] analyzed 45 studies about HS as a predictor of adverse mortality outcomes and functional decline [12,25]. Furthermore, muscle strength seems to be a reliable indicator of health associated with cardiovascular risk and metabolic syndrome in adults and elders [27,28]. In this context, Sayer et al. [29] described an association between the decrease in HS and a suboptimal score in general health, physical activity, and mental health. Altogether, HS seems to be a good predictor of life quality related to physical and mental health status [30], including, at the same time, social interactions between an individual and the environment [31]. For all these reasons, HS was an appropriate test to assess physical condition and fatigue in nurses.

Taking this information into account, the HS test is related to the overall strength that, at the same time, is related to the level of fatigue and stress [23]. In the present study, as expected, women had lower strength than men, and the percentage of decrease at T2 was more pronounced in women than in men (Table 2). The data reflect a certain state of fatigue in both genders, particularly in women at T2. In addition, women are the dominant gender of nurses in the Spanish Health system. Fatigue is confirmed by increased blood lactate levels determined after the HS test at T2. The strength measured in this test at T2 was lower than the strength determined at T1. However, cortisol data show that fatigue is not accompanied by stress at T2. Otherwise said, nurses were more stressed at the beginning of the working day than at the end. A likely interpretation is that nurses are adapting throughout the working shift to the daily situation. An additional point to consider is that women seem to be more sensitive to stress than men. In women, the correlation chart (Table 6) indicates that cortisol correlates modestly and positively to strength and modestly but negatively to lactate at T1 and T2. A likely interpretation may suggest that the physical effort (HS test) is performed under stress at T1, but this situation changes at T2, where the effort is performed under less stress but with increased fatigue. Nevertheless, correlations are modest, and this interpretation needs additional research.

Regarding QPL, dimensions one and three (“Managerial support” and “Intrinsic motivation”, respectively) displayed very similar high scores in the CVP-35 questionnaire (Table 4). However, as observed previously [21], dimension two (“Workloads”) gives the lowest scores, conditioning the score of “Quality of life at work” (Table 4). This situation is probably caused by the demanding tasks of the ED, giving a suboptimal recovery. Once ED nurses have started their work (higher cortisol at the beginning of the shift), they adapt to the situation and are ultimately not influenced by the organizational and work component of the ED. In any case, the results of the present report are similar to others referring to the general QPL [8,32]. In this context, the CVP-35 questionnaire seems to be a reliable instrument, easy to apply, and based on an established conceptual framework with a stable pattern of response in different environments [32]. Otherwise said, CVP-35 is a useful test for the measurement of QPL, giving at the same time the possibility to establish comparisons between different organizations and places, i.e., EDs vs. other Hospital Departments. In this context, QPL integrates the satisfaction, health, and well-being of the worker and everything related to the working environment. In this line, optimal QPL has to reach a balance between the demands of the job and the perceived ability to cope with them, including all those job conditions that can affect motivation and performance [11,33]. When an imbalance appears, this leads to job stress affecting job satisfaction and the perception of optimal QPL. This situation contributes to sleep disturbances and anxiety, increasing stress. In this regard, it was previously reported [34] that ED professionals working in a pre-pandemic situation had adequate work management, with good sleep quality, low anxiety, and high efficiency, controlling stress during professional performance.

Finally, the tests were carried out on a normal day, without any special circumstances requiring greater attention and precision in their professional work. Future research on a more demanding day in an ED could give interesting results. Anyway, the study was performed during the pre-pandemic period, which is the usual working organization in Spanish Health System. Pandemics was an unusual period, and comparisons are not the objective of the present study. Lastly, we selected only nursing staff working only in the morning shift. The situation of another professional working in an ED will be considered in future research. In this line, shift work may contribute to an increase in the number of illnesses and increase the number of medical errors each year [35], and all professionals working in EDs could be affected. In this regard, nurses working shifts longer than 12.5 h experienced decreased vigilance and occupational injuries, such as a needle stick [36]. Similarly, 24 h shift workers were 36% more prone to make a serious medical error and 300% more predisposed to make a fatigue-related error that could result in patient death compared to their 16 h shift colleagues [35]. Folkard and Tucker [37] have studied the interactions between human capabilities, work demands, and working hours and the potential consequences for safety and productivity. These authors indicate that minimizing overall risk in a shift system requires considering the number of successive night shifts, the length of these shifts, and the provision of breaks within shifts.

## 5. Conclusions

In conclusion, the present study reports low values of stress (low cortisol at T2) in ED nurses due to a positive adaptation to working demands during the shift. This adaptation is accompanied by an acceptable level of QPL going, together with the development of fatigue at T2 (increased lactate and less strength). Therefore, “Workloads” is a variable that needs optimization to improve overall QPL. In this line, it would be interesting to implement measures leading to a better organization of work through ergonomic, safe, and healthy conditions, giving a response to the needs and demands of workers and increasing, if necessary, the number of workers in ED.

## Figures and Tables

**Figure 1 ijerph-20-02052-f001:**
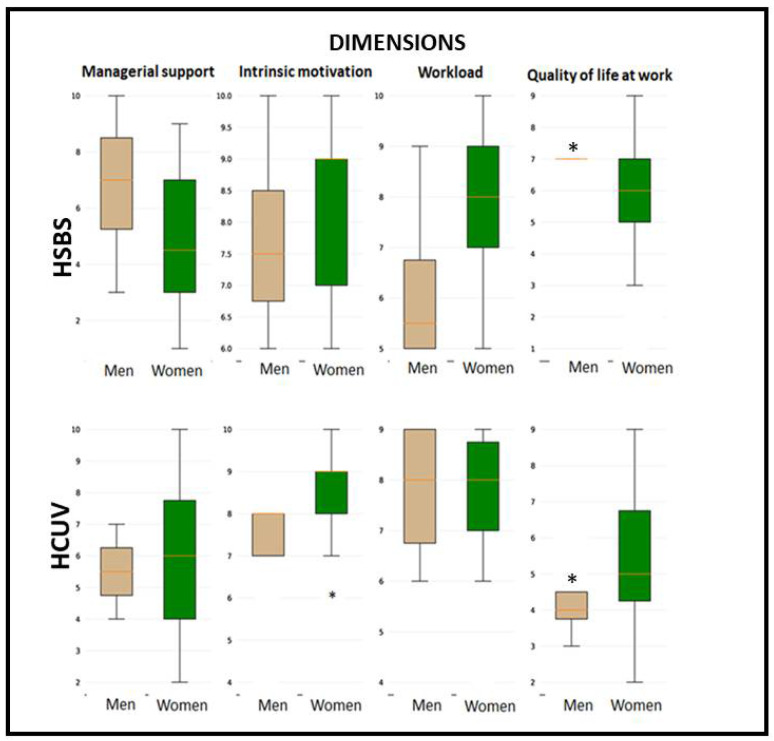
Overall median score for each dimension of the CVP-35 questionnaire in nursing staff in both hospitals (HSBS and HCUV), separated by gender. The 95% confidence interval of the median, calculated by bootstrap, is shown. * Indicates significant differences between median values for men and women, (Krustall-Wallis *p* < 0.05).

**Table 1 ijerph-20-02052-t001:** Demographic data of the nurses participating in the study.

HOSPITALS	n	Gender
Total	59	49 Women/10 Men
HCUV	28	23 Women/5 Men
HSBS	31	26 Women/5 Men
**AGE (years)**		
HCUV		
Nurses (Women)	35.6 ± 10.6	
Nurses (Men)	34.6 ± 7.0	
HSBS		
Nurses (Women)	42.6 ± 13.1	
Nurses (Men)	45.2 ± 15.7	

Note: HCUV: “Hospital Clínico Universitario” of Valladolid; HSBS:“Hospital Santa Bárbara” of Soria.

**Table 2 ijerph-20-02052-t002:** Dynamometry values and post-test blood lactate levels in nurses (men and women) from both hospitals. The 95% confidence interval of the median, calculated by bootstrap, is shown.

HSBS	WOMEN	MEN
Dynamometry T1 (Kg)	29.0 ± 1.0	43.0 ± 9.0 *
Dynamometry T2 (Kg)	27.5 ± 1.5	44.0 ± 8.0 *
Δ% Change	−5.17	2.33
Lactate T1 (mmol/L)	2.2 ± 0.5	2.6 ± 1.4
Lactate T2 (mmol/L)	3.2 ± 0.8 ^#^	5.5 ± 2.3 *^,#^
Δ% Change	51.2	115.7
**HCUV**	**WOMEN**	**MEN**
Dynamometry T1 (Kg)	27.0 ± 3.0	44.0 ± 12.0 *
Dynamometry T2 (Kg)	24.5 ± 3.5	41.0 ± 11.0 *
Δ% Change	−9.3	−6.8
Lactate T1 (mmol/L)	2.0 ± 0.2	2.1 ± 0.5
Lactate T2 (mmol/L)	2.9 ± 0.4 ^#^	2.5 ± 1.1^#^
Δ% Change	45.0	19.5

The ∆% change was calculated comparing the T2 values to T1 (considered as 100%). * Significant differences between median values comparing men vs. women (Krustall-Wallis *p*-value < 0.05). ^#^ Significant differences between T1 vs. T2.

**Table 3 ijerph-20-02052-t003:** Salivary cortisol, in men and women from both hospitals. The 95% confidence interval of the median, calculated by bootstrap, is shown.

HSBS	WOMEN	MEN
Cortisol T1 (µg/dL)	10.1 ± 3.5	14.7 ± 10.1
Cortisol T2 (µg/dL)	1.3 ± 0.3 ^#^	1.5 ± 0.8 ^#^
Δ% Change	−87.0	−90.0
**HCUV**	**WOMEN**	**MEN**
Cortisol T1 (µg/dL)	7.4 ± 1.9	23.4 ± 14.4 *
Cortisol T2 (µg/dL)	1.2 ± 0.3 ^#^	3.0 ± 2.4 ^#^
Δ% Change	−84.0	−87.1

The ∆% change was calculated comparing the T2 values to T1 (considered as 100%). * Significant differences between median values comparing men vs. women (Krustall-Wallis p-value< 0.05). ^#^ Significant differences between T1 vs. T2.

**Table 4 ijerph-20-02052-t004:** Overall median score for each dimension of the CVP-35 questionnaire in the nursing staff in both hospitals (HSBS and HCUV). The 95% confidence interval (CI) of the median, calculated by bootstrap, is shown.

Factors	95% Median CI	
	HSBS	HCUV
**Managerial support factor (dimension 1)**	7.3 ± 1.3	7.2 ± 1.3
**Workload factor (dimension 2)**	6.0 ± 1.1	6.2 ± 1.0
**Intrinsic motivation factor (dimension 3)**	8.3 ± 0.8	8.2 ± 0.7
**Quality of life at work**	6.1 ± 1.0	6.0 ± 1.2

**Table 5 ijerph-20-02052-t005:** Correlation chart in men comparing different variables (T1: star of study, T2: end of study).

		Dynamometry (Kg)	Dynamometry (Kg)	Lactate (mmol/L)	Lactate (mmol/L)	Cortisol (μg/dL)	Cortisol (μg/dL)
		T1	T2	T1	T2	T1	T2
**Dynamometry (Kg)**	**T1**	1	0.84	−0.061	−0.005	0.014	0.028
**Dynamometry (Kg)**	**T2**	0.84	1	0.03	0.067	−0.037	−0.025
**Lactate (mmol/L)**	**T1**	−0.061	0.03	1	0.26	−0.0072	−0.0087
**Lactate (mmol/L)**	**T2**	−0.005	0.067	0.26	1	0.14	0.022
**Cortisol (μg/dL)**	**T1**	0.014	−0.037	−0.0072	0.14	1	0.11
**Cortisol (μg/dL)**	**T2**	0.028	−0.025	−0.0087	0.022	0.11	1

**Table 6 ijerph-20-02052-t006:** Correlation chart in women comparing different variables.

		Dynamometry (Kg)	Dynamometry (Kg)	Lactate (mmol/L)	Lactate (mmol/L)	Cortisol (μg/dL)	Cortisol (μg/dL)
		T1	T2	T1	T2	T1	T2
**Dynamometry (Kg)**	**T1**	1	0.94	−0.11	0.14	0.32	0.29
**Dynamometry (Kg)**	**T2**	0.94	1	0.21	−0.061	0.38	0.2
**Lactate (mmol/L)**	**T1**	−0.11	0.21	1	0.48	0.28	−0.34
**Lactate (mmol/L)**	**T2**	−0.14	−0.061	0.48	1	0.19	−0.34
**Cortisol (μg/dL)**	**T1**	0.32	0.38	0.28	0.19	1	0.68
**Cortisol (μg/dL)**	**T2**	0.29	0.2	−0.34	−0.34	0.68	1

## Data Availability

Data are available from the corresponding author upon reasonable request.

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
