# Peer review of "Relationship between Strength and Professional Quality of Life of Nurses Working Hospital Emergency Departments"

_ijerph, 2023, doi:10.3390/ijerph20032052_

Round 1

Reviewer 1 Report

Dear author/authors

I read your manuscript. Your project is good, but I think it needs to improve somewhere. Sentences must be written passively. What questions the questionnaire consists of?  The sections of result and discussion are fairly poor. In the result, you could assess more factors among nurses. You could assess the relation between more variables and could assess correlation between questions and Cortisol, Lactate and Hs tests and could write better discussion.

Figures 2 and 3 seems to be Tables.

In the discussion, I didn't see the strong discussion.

Author Response

ANSWER ABOUT COMMENTS OF REVIEWER-1

I read your manuscript. Your project is good, but I think it needs to improve somewhere. Sentences must be written passively.

ANSWER-1: The manuscript is written in passive. Nevertheless, sentences of lines 76, 173, 179, 183, 221, 254 and 305 were changed to the passive mode.

What questions the questionnaire consists of?

ANSWER-2: The QPL/CVP-35 was extensively described in lines 138-150. The CVP-35 is based in Reference 20. The validated Spanish version is described in References 4 and 5. For this reason, questions are in Spanish but we have translated. The questions are shown in Supplementary Material.

The sections of result and discussion are fairly poor. In the result, you could assess more factors among nurses. You could assess the relation between more variables and could assess correlation between questions and Cortisol, Lactate and HS tests and could write better discussion.

ANSWER-3: Dear reviewer, we think that these correlations that you propose do not contribute to anything new. Answers from the CVP-35 questions do not bring specific parameters more significance than the others. The instrument used was the CVP-35, which measures professional quality of life in a general way. The questions are grouped into three dimensions: "managerial support", "work demands" and "intrinsic motivation", although not all authors group them into the same dimensions. For this reason, and having analyzed this proposal, we believe that it will not be able to bring us anything new.

Figures 2 and 3 seems to be Tables.

ANSWER-4: Figures 2 and 3 are colored correlation charts. These figures do not follow the format of the Tables according to the journal instructions. For this reason, they are considered as figures in the first version of the manuscript. Nevertheless, Reviewer-2 recommend as well to convert Figures 2 and 3 into Tables. See the new Tables 5 and 6.

In the discussion, I didn't see the strong discussion.

ANSWER-5: Discussion has been changed accordingly. 

Reviewer 2 Report

Title: Relationship between Strength and Professional Quality of Life of Nurses Working Hospital Emergency Departments

The manuscript requires thorough corrections.

  1. The introduction is too long...
  2. The authors did not specify when exactly they performed their research.
  3. The authors of the study examined a very small number of participants, only 59 nurses.

There is no describe study design.

They conducted their research only during one shift. You make comparisons between the genders. What about other socio demographics data, like age, places of residence, education?

Conclusions cannot be made.

The authors should improve their research.

In Materials and methods you write : <<„A total of 59 nurses (10 men and 49 119 women) were recruited for the study”>> but in results: <<„We observed that nurses manifested a  lower “Managerial Support” (dimension 1 in CVP-35) than Medicine Doctors. “Workload” (dimension 2 in CVP-35) perception was very similar between nurses and Medicine Doctors. However, “Intrinsic Motivation” (dimension 3 in CVP-35) was higher in nurses  than in Medicine Doctors. Altogether, the overall perceived QPL is higher in nurses than  in Medicine Doctors, but still low.”>>. My question is: Who are the participants in the study?

Figures 1., 2. and 3. should be converted into tables.

  1. Limitations of the work should result from the conducted research.
  2. You wrote: „In conclusion, the present study reports low values of stress …”. The conclusion is written too quickly. Only 59 nurses were included in the study. Only nurses? or nurses and medical doctors?
  1. References should be more up to date.

Author Response

REVIEWER-2

Title: Relationship between Strength and Professional Quality of Life of Nurses Working Hospital Emergency Departments

The manuscript requires thorough corrections.

  1. The introduction is too long...

ANSWER: Introduction has been shortened accordingly (see deleted lines).

  1. The authors did not specify when exactly they performed their research.

ANSWER: The research was performed before pandemics in 2019. This was indicated in the first version of the manuscript (first line in section “Participants”).

  1. The authors of the study examined a very small number of participants, only 59 nurses.

ANSWER: Similar numbers were analyzed in previous reports where we calculated sample size. Therefore, this number is representative of the nurses’ population in Emergency Departments in Spain. See References 18, 20 and 34.

There is no describe study design.

ANSWER: Analytical descriptive cross-sectional study. This was indicated in the first version of the manuscript (first line in section “Participants”).

They conducted their research only during one shift. You make comparisons between the genders. What about other socio demographics data, like age, places of residence, education?

ANSWER: We agree that we can do a lot of comparisons and correlations, but gender is a main social and political concern in the present time. For this reason, we gave priority to this variable. The rest of variables are very interesting and we plan to study in future reports.

Conclusions cannot be made.

ANSWER: Conclusions are a section of the manuscript according to Journal instructions. Therefore, we cannot eliminate.

The authors should improve their research.

ANSWER: This manuscript represents a part of a main research project (see References 18, 20 and 34). We have still points to analyze in future research. In any case, the concerns raised by the Reviewer-2 will improve the final quality of the present manuscript.

In Materials and methods you write: <<„A total of 59 nurses (10 men and 49 women) were recruited for the study”>> but in results: <<„We observed that nurses manifested a lower “Managerial Support” (dimension 1 in CVP-35) than Medicine Doctors. “Workload” (dimension 2 in CVP-35) perception was very similar between nurses and Medicine Doctors. However, “Intrinsic Motivation” (dimension 3 in CVP-35) was higher in nurses than in Medicine Doctors. Altogether, the overall perceived QPL is higher in nurses than in Medicine Doctors, but still low.”>>. My question is: Who are the participants in the study?

ANSWER: Nurses are participants in the study. We compare the results obtained in nurses with Doctors, because they are working in the same environment, but with different responsibilities. During our research we analyzed both populations at the same time (see References 18, 20 and 34). However, the obtained results prompted us to go in more detail in the group of nurses (present report).

Figures 1., 2. and 3. should be converted into tables.

ANSWER: Figures 2 and 3 has been converted into Tables 5 and 6.  We think Figure 1 is more expressive as it is now. The effect would not be evident if the Figure is transformed into a Table.

  1. Limitations of the work should result from the conducted research.

ANSWER: We agree with the Reviewer that making the study during out of the pandemics period could not be considered as a limitation. The normal situation in the Spanish Health system has been the no pandemics situation. The system is coming actually to this normality and this is the situation that Health administrators have to face and improve. Moreover, an additional research needs to be carried out in a day with emergency services. For this reason, this is not a limitation, but a different situation to investigate and get additional information. For all these reasons, we have changed the sentences regarding limitations of the study and we have considered no limitations.

  1. You wrote: „In conclusion, the present study reports low values of stress …”. The conclusion is written too quickly. Only 59 nurses were included in the study. Only nurses? or nurses and medical doctors?

ANSWER: The study was performed only in nurses. As indicated before, Medical Doctors and other professionals working in ED were studied in a future research.

  1. References should be more up to date.

ANSWER: No much research was carried out in this field. For this reason, References are appropriate.